# Effects of a Personalized Diet on Nutritional Status and Renal Function Outcome in Nephrectomized Patients with Renal Cancer

**DOI:** 10.3390/nu16091386

**Published:** 2024-05-03

**Authors:** Francesco Trevisani, Fabiana Laurenti, Francesco Fiorio, Matteo Paccagnella, Matteo Floris, Umberto Capitanio, Michele Ghidini, Ornella Garrone, Andrea Abbona, Andrea Salonia, Francesco Montorsi, Arianna Bettiga

**Affiliations:** 1Division of Experimental Oncology, Urological Research Institute, IRCCS San Raffaele Scientific Institute, 20132 Milan, Italy; fiorio.francesco@hsr.it (F.F.); capitanio.umberto@hsr.it (U.C.); salonia.andrea@hsr.it (A.S.); montorsi.francesco@hsr.it (F.M.); 2Department of Urology, IRCCS San Raffaele Scientific Institute, Vita-Salute San Raffaele University, 20132 Milan, Italy; 3Department of Medicine and Surgery, University of Parma, 43126 Parma, Italy; fabi.laurenti@gmail.com; 4Translational Oncology Fondazione Arco Cuneo, 12100 Cuneo, Italy; matteo.babeuf@gmail.com (M.P.); abbona.andrea@gmail.com (A.A.); 5Department of Nephrology, Dialysis, and Transplantation, G. Brotzu Hospital, 09134 Cagliari, Italy; matteo.floris@aob.it; 6Department of Oncology, Fondazione IRCCS Ca’ Granda Ospedale Maggiore Policlinico, 20122 Milan, Italy; michele.ghidini@policlinico.mi.it (M.G.); ornella.garrone@policlinico.mi.it (O.G.); 7Faculty of Medicine and Surgery, Vita-Salute San Raffaele University, 20132 Milano, Italy

**Keywords:** nutrition, kidney cancer, onco-nephrology, CKD, diet

## Abstract

Nutritional therapy (NT) based on a controlled protein intake represents a cornerstone in managing chronic kidney disease (CKD). However, if a CKD patient is at the same time affected by cancer, oncologists and nutritionists tend to suggest a dietary regimen based on high protein intake to avoid catabolism and malnutrition. International guidelines are not clear when we consider onco-nephrological patients and, as a consequence, no clinical shared strategy is currently applied in clinical practice. In particular, no precise nutritional management is established in nephrectomized patients for renal cell carcinoma (RCC), a specific oncological cohort of patients whose sudden kidney removal forces the remnant one to start a compensatory mechanism of adaptive hyperfiltration. Our study aimed to investigate the efficacy of a low–normal-protein high-calorie (LNPHC) diet based on a Mediterranean model in a consecutive cohort of nephrectomized RCC patients using an integrated nephrologist and nutritionist approach. A consecutive cohort of 40 nephrectomized RCC adult (age > 18) patients who were screened for malnutrition (malnutrition screening tool, MST < 2) were enrolled in a tertiary institution between 2020 and 2022 after signing a specific informed consent form. Each patient underwent an initial nephrological and nutritional evaluation and was subsequently subjected to a conventional CKD LNPHC diet integrated with aproteic foods (0.8 g/Kg/die: calories: 30–35 kcal per kg body weight/die) for a period of 6 months (±2 months). The diet was structured after considering eGFR (CKD-EPI 2021 creatinine formula), comorbidities, and nutritional status. MST, body mass index (BMI), phase angle (PA), fat mass percentage (FM%), fat-free mass index (FFMI), body cell mass index (BCMI), extracellular/intracellular water ratio (ECW/ICW), extracellular matrix/body cell mass ratio (ECM/BCM), waist/hip circumference ratio (WHC), lab test exams, and clinical variables were examined at baseline and after the study period. Our results clearly highlighted that the LNPHC diet was able to significantly improve several nutritional parameters, avoiding malnutrition and catabolism. In particular, the LNPHC diet preserved the BCM index (delta on median, ΔM + 0.3 kg/m^2^) and reduced the ECM/BCM ratio (ΔM − 0.03 *), with a significant reduction in the ECW/ICW ratio (ΔM − 0.02 *), all while increasing TBW (ΔM + 2.3% *). The LNPHC diet was able to preserve FFM while simultaneously depleting FM and, moreover, it led to a significant reduction in urea (ΔM − 11 mg/dL **). In conclusion, the LNPHC diet represents a new important therapeutic strategy that should be considered when treating onco-nephrological patients with solitary kidney due to renal cancer.

## 1. Introduction

In the last decade, onco-nephrology arose as a prominent new subspeciality of nephrology due to the bidirectional relationship between kidney disease and cancer, known as the “the kidney–cancer connection”. Nowadays, new lines of evidence demonstrate that the development of malignancies in patients results in renal dysfunction, derived both from systemic cancerous biological modifications (e.g., inflammation) and the nephrotoxic agents administered by clinicians during treatment [1]. Therefore, the onset of acute kidney injury (AKI), acute kidney disease (AKD), and chronic kidney disease (CKD) in onco-nephrological patients has become a new insidious clinical threat that requires innovative therapeutic strategies to avoid end-stage renal disease (ESRD) and subsequent dialysis in such a cohort of patients [2,3].

Among these new lines of nephrological therapies, establishing a precise, tailored diet that could slow the progression of renal damage in acute and chronic conditions has only been partially considered in daily clinical practice [4,5]. The role of a controlled protein diet in CKD management has been well elucidated, with clear benefits for renal function [6]. In the same manner, ESPEN guidelines for oncological patients outline a clear protective role of a high protein intake against lean mass depletion and cachexia [7]. This being considered, we see the first incongruence between two guidelines, which arises from the fact that these pathologies are never talked about in combination; in fact, ESPEN guidelines never mention specific indications for disease management in nephropathic oncological patients. This is of great importance, since it is known that a high protein intake (namely > 1.2 g/kg) affects renal hemodynamics, increasing renal blood flow, intraglomerular pressure, and glomerular filtration rate [8]. In this context, therefore, the introduction of a low–normal-protein high-calorie (LNPHC) diet in cancer patients affected by CKD is still debatable due to the risk of catabolism and malnutrition [9], and the fact that, since other international guidelines are not directive in this area of patients, no nutritionally shared strategy is present at all [10,11,12]. While individuals with healthy intact kidneys may not be affected by the harmful impact of a high-protein diet, those with limited nephron endowment and at risk of CKD may be more vulnerable, such as diabetic and obese people, as well as those with reduced kidney reserve such as solitary kidney or the earlier stages of CKD [13].

Among the onco-nephrological patient subcategories, there is, in fact, one in particular that deserves an immediate, clear nutritional strategy to avoid false evaluations and worse clinical outcomes: renal cell cancer patients (RCC) who are left with a solitary kidney after nephrectomy [14,15]. This cohort of patients experiences the sudden removal of a kidney, resulting in the loss of half of their renal endowment [16,17]. As a consequence, the development of renal dysfunction due to the surgical removal of nephron mass implicates a proper multidisciplinary approach between urologists, oncologists, and nephrologists with immediate effect in order to maintain the correct clinical balance to avoid the severe stages of AKI, AKD, and CKD on one side, and to treat cancer on the other [18,19,20]. However, few reports in the literature are present in terms of nutritional regimen in this scenario because specialists usually tend to give prominence to clinical aspects as opposed to diet and, as a consequence, some patients are treated with high protein intakes, some with low protein quotas, and others with a self-determined diet [21,22,23]. 

All things considered, our study aimed to investigate the role of an LNPHC diet in the clinical management of nephrectomized RCC patients using a combined nephrological and nutritional approach focusing on both the clinical aspect of disease management and on the impact on body composition, with the objective of avoiding malnutrition while decreasing the burden on the contralateral kidney.

## 2. Materials and Methods

An observational retrospective study was conducted in a cohort of 40 consecutive patients who underwent radical nephrectomy for renal cancer in a tertiary-care institution (IRCCS San Raffaele Hospital, Milan, Italy) between 2020 and 2022, and who were assigned a low–normal-protein high-calorie (LNPHC) diet for a period of 6 ± 2 months following an initial multidisciplinary evaluation (by an oncologist, nephrologist, and nutritionist). 

The timing of patient enrollment started from about 3 months after surgery in order to consider stable renal function after acute nephron mass removal, avoiding misleading evaluations due to transient values of GFR. This decision is in line with international nephrological guidelines where the establishment of reliable chronic renal function follows after a period of 90 days after the acute renal event (in this case, the nephrectomy). This period of 3 months is defined as acute kidney disease (AKD), and it represents a transient period of renal adaptation to the new kidney status. After this period, nephrologists can determine whether the obtained GFR is a chronic GFR, and therefore define patients as either having normal renal function or chronic kidney disease. 

The exclusion criteria for cohort selection were age < 18 years, renal function not stable for three months, end-stage renal disease requiring hemodialysis, concomitant chemotherapy or immunotherapy, metastasis, and the absence of informed consent. General health status was clinically recorded at each visit for each patient, taking into account clinical condition (which included history of hypertension, diabetes, and medical therapy (ACE inhibitors (ACEi), angiotensin II receptor blockers (ARBs), calcium antagonists, beta blockers, and diuretics)), weight, blood pressure, the presence of edema, and bioimpedance analysis (BIA). Patients were included in a nutrition education program which consisted of an individualized diet plan based on the patient’s initial nutritional status and renal function. This program included an interview intended to capture detailed information about the foods and beverages usually consumed, 2 nutrition education sessions, a nutritional assessment, and monitoring for 6 ± 2 months. Both the 24 h recalls (used to estimate usual food intake) and the dietary intervention were performed by a single nutritionist with the aim of providing a personalized dietary prescription based on the main recommendations of the Mediterranean diet, with some quantitative changes introduced to the supply of nutrients to adapt the diet to the patient’s clinical needs [24]. The main features of the diet prescription included limitations to the amount of protein (0.8 g/kg/day, reserving a quantity of protein < 0.8 g/day only for patients with CKD-IV if necessary), salt (6 g/day), and phosphate (800 mg/day).

The daily energy intake ranged from 30 to 35 kcal/kg/day, with the suggestion to move more and participate in any amount of moderate-to-vigorous physical activity to contrast calorie intake [25], and protein-free commercial foods were introduced into the diet when needed. Patients requiring specific interventions to address deficiencies, such as vitamin D, calcium, folic acid, vitamin B12, or others, received appropriate supplementation according to established good clinical practice criteria [1].

Laboratory tests were gathered at baseline and the end of the study following internal routine blood exams. The glomerular filtration rate (GFR) was estimated at each time point using the creatinine-based estimated glomerular filtration rate (eGFR) formula CKD-EPI (2021) [26]. The CKD categories were assigned according to the KDIGO guidelines [3]. Gold-standard measurements using Iohexol plasma clearance were employed to eliminate possible bias linked with sarcopenia for eGFR [4].

The study received the approval of the Institutional Ethical Committee (San Raffaele Hospital, Milan, Protocol Code/Acronym URBBAN, approval date 3 March 2014), and informed consent was obtained from all of the patients included in the study. All the experimental procedures involving human biological material complied with the approved guidelines and were in accordance with good clinical practice. 

### 2.1. Assessment of Nutritional Status

Anthropometric measurements were performed to assess each patient’s nutritional status. We collected body weight and height for each patient to calculate the corresponding body mass index (BMI), waist and hip circumference and their ratio, and triceps skinfold thickness (TSF). Bioelectrical impedance analysis (BIA) was used to study the patients’ body composition: we collected data for phase angle (PA), body cellular mass/height^2^ ratio (BCM/h^2^), extracellular mass/BCM ratio (ECM/BCM), extracellular water/intracellular water ratio (ECW/ICW), total water (TBW%), fat mass/height^2^ ratio (FM/h^2^), fat-free mass/height^2^ ratio (FFM/h^2^ or FFMI), and mid-upper-arm muscle circumference (MAMC). MAMC was calculated using the following formula: MAMC = mid-arm circumference − (3.1415 × TSF). The Malnutrition Universal Screening Tool (MUST) was used to identify malnutrition.

### 2.2. Statistical Analysis

Comparisons among values of each variable at T1 and T0 were performed using the Mann–Whitney U test, or the Wilcoxon signed-rank test was performed for paired samples. Normal distribution was analyzed using Student’s *t*-test or a paired *t*-test. The deltas of ECM/BCM and ICW/ECW between T1 and T0 were computed as dependent variables, with patients who experienced a reduction in eGFR more significant than 10% as a fixed factor and creatinine as the confounding factor in the ANCOVA analysis. Differences in categorical variables were analyzed with a χ^2^ test or Fisher’s exact test.

The Mann–Whitney U test, Student’s *t*-test, Wilcoxon signed-rank test, χ^2^ test, and Fisher’s exact test were performed with GraphPad v.5 (GraphPad Software, Boston, MA, USA).

ANCOVA was assessed using SPSS V.24 (IBM SPSS Statistics for Windows, Version 24.0. Armonk, NY, USA).

In all tests, *p* < 0.05 was regarded as significant. The Benjamini–Hochberg (B-H) procedure was applied to decrease the false discovery rate by 10% [27].

## 3. Results

### 3.1. Clinical and Demographic Findings of the Patients 

Table 1 shows the characteristics of forty consecutive patients at baseline before the nutritional and nephrological combined treatment. The median age was 64, the most predictable age for RC development, with a predominance of males (80%) over females in line with RCC epidemiology. Regarding comorbidities, 22 (55%) patients had hypertension, whereas 7 (17.5%) patients had type II diabetes. After collecting BMI at baseline with a median of 25.8 (23.1–28.7), we found that 80% of the subjects had average or slightly high body weight, while only 20% had an excessive body weight. Among our cohort, none of the patients were underweight. In terms of renal function, no patient was at CKD stage V, 10 patients (25%) displayed CKD stage IV, 23 were at CKD stage III (59%), 6 were at CKD stage II (15%), and only 1 was at CKD stage I (2.5%). It is essential to underline that the grade of renal function was obtained using serum creatinine and estimated GFR values after radical nephrectomy, i.e., with “half” global kidney mass. 

Following the Mediterranean diet and normalizing protein intake to the RDA value of 0.8 g/kg/day represents a protective measure for the general population; the benefit is even more pronounced for the majority of patients at risk of CKD, as well as persons with obesity, hypertension, diabetes mellitus, or solitary kidney. Data from the National Health and Nutrition Examination Survey (NHANES) show that the average consumption of protein in adults with and without CKD in the United States is estimated to be approximately 1.2–1.4 g/kg per day, which is higher than the recommended amount [28]. The same trend has been seen for European countries such as Italy (https://knowledge4policy.ec.europa.eu/health-promotion-knowledge-gateway/dietary-protein-overview-countries-6_en (accessed on 10 March 2024); [29]). Hence, a careful evaluation of dietary habits represents the first step of any nutritional intervention and gives the qualitative and quantitative information needed to develop targeted counseling. In this regard, we used 24 h recalls to estimate habitual energy and nutrient intake and to better understand the patients’ lifestyles. From the evaluation of the 24 h recall diaries, it emerged that, in order to meet the Mediterranean diet targets, the consumption of fruits, vegetables, legumes, and cereals needed to be substantially increased in our population; conversely, the consumption of animal protein in the form of meat (red and white), fish, cured meats, and aged cheeses needed to be halved. Our evaluation of the diaries showed a similar trend to what was observed in population-based studies performed with the use of food frequency questionnaires: our cohort’s nutritional habits were characterized by a high intake of meat and a low intake of vegetables, fruits, and legumes, as observed in the adult Italian population [30,31,32].

### 3.2. Overall Improvement in Anthropometric Indices following the Nephrological–Nutritional Combined Approach of an LNPHC Regimen for Nephrectomized Patients

Table 2 collects the anthropometric indices of our patients before and after dietary treatment. The table shows that our study’s most prevalent nutritional status was overweight–obese, as defined by the BMI category; however, this definition could be different if body composition was measured directly through distinct techniques. For these reasons, we used other indices obtained from two separate indirect methods of body composition estimation that are relatively easy to perform: skinfold thickness measurement and body impedance analysis (BIA). Our data showed that, for the same BMI, the dietetic treatment determined a significant reduction both in the surrogate measurement of abdominal obesity (waist circumference) and in segmental fat distribution (TSF). 

In parallel, when focusing on the BIA parameters, this table underlines that our cohort of patients presented an adequate FFM value that was not affected by the dietary approach, meaning the treatment used was effective in preserving the patients’ nutritional status.

Furthermore, we noticed a stable BCM index and a significant decrease in the ECM/BCM ratio, and these observations confirmed not only the absence of malnutrition insurgence in the cohort, but also an improvement in healthy fluid balance. It is worth remembering that ECW and ICW in skeletal muscle reflect the extracellular space filled with plasma and interstitial fluid and the muscle cell mass (i.e., myofibers), respectively [33]. The ECW/ICW ratio improved significantly, as did the TBW%, which followed an upward trend; these observations further support that the treatment did not deplete muscle mass, but improved nutritional status overall. Finally, we must point out a significant increase in MAMC in our population. This result will be elaborated further in the discussion. 

### 3.3. Nephrological Clinical Parameters before and after LNPHC Diet

Table 3 and Figure 1 show the results regarding the variation in the most relevant nephrological clinical parameters before and after the dietary regimen, which are serum creatinine and urea values together with eGFR. It is important to underline that, due to using the eGFR formula (instead of a measured GFR tool such as Iohexol or iothalamate plasma clearance), which is based on serum creatinine levels, mild fluctuations in both parameters could be defined as physiological after a radical nephrectomy, and therefore not statistically significant. The table highlights an important and significant reduction in urea blood levels (*p* value: 0.0006) after the dietary regimen, suggesting that the LNPHC diet was fundamental in improving this dangerous nitrogenous end product of metabolism in each patient. Specifically, individuals with initial urea levels exceeding 50 mg/dL experienced a notable improvement following the dietary regimen. It is essential to emphasize that, in patients with homologous characteristics to our cohort, a high quantity of serum urea can dramatically compromise cognitive assets with a concomitant worsening of pre-existing renal impairment and an augmented risk of end-stage kidney disease requiring dialysis [34]. In addition, the significant reduction in urea could suggest that no patient developed protein catabolism due to malnutrition with the reduction in protein intake, or else the levels of urea would have increased.

### 3.4. Statistical Correlations between Delta eGFR Values and Anthropometric Indices Using BIA

Our data confirmed a positive correlation between FMI and ECW/ICW, as well as ECM/BCM, as showed in Table 4. Our data highlight that the dietary treatment resulted in a significant reduction in FMI, which is correlated with a decrease in both ECW/ICW and ECM/BCM. Therefore, the diet was able to reduce both the expansion of the extracellular matrix while keeping BCMI constant, and improve the water balance without significant increases in the FFMI.

Considering that the dietary treatment did not show significant changes in the FFMI and that the latter can be considered an indicator of nutritional status, a negative correlation between total water and the ratios of ECW/ICW and ECM/BCM confirms that stable FFMI values are due to the preservation of the cellular component, and not to the expansion of the extracellular water component or edema. This is also confirmed by the significant increase in MAMC post-treatment.

The relationship between the 10% worsening of the eGFR and delta (T1-T0) ECW/ICW was evaluated using analysis of covariance (ANCOVA) with an appropriate contrast (linearity). Similarly, the relationship between delta (T1-T0) ECM/BCM and the 10% worsening of the eGFR was analyzed. The value of creatinine at T0 was used as a covariate in these models. The data in Table 5 showed a significant difference in the mean ECW/ICW variation between pre- and post-treatment and the patients presenting a 10% worsening in eGFR. Specifically, patients who had a worsened eGFR showed an increase in ECW/ICW. We cannot say the same for the variation in ECM/BCM. The presence or absence of ECW expansion upon physical examination may be useful in predicting a functional decline in CKD patients.

## 4. Discussion

Renal cancer represents the 10th most common malignancy globally, and it is increasing worldwide due to its links to obesity and diabetes [10]. Radical or partial nephrectomies represent the first line of therapy to treat the disease, even in metastatic phases [10]. However, the detrimental impact on renal function, both in acute and chronic cases, as a result of this surgical approach can dramatically compromise the lifespan of cancer patients with the onset of both AKI and CKD [19]. Therefore, in the last decade, urologists have ameliorated their surgical techniques by performing nephron-sparing methods on small renal masses to preserve as many kidneys as possible [14,35]. Unfortunately, RC is frequently clinically silent until its advanced stages because the disease is characterized by several specific symptoms that clinicians do not easily recognize as cancer side effects. Subsequently, a non-negligible proportion of patients discover RC when the disease has already compromised the entire kidney, and the only strategy to remove the malignancy is radical nephrectomy [36]. Because the goal of cancer management is no longer to simply treat the disease, but also to reduce the risk of further morbidity and mortality, the onset of chronic renal damage remains an insidious threat for both clinicians and patients because it continuously affects the lives of oncological patients, both when the cancer is completely eradicated and when a metastatic process is ongoing [18]. For the abovementioned reasons, nephrectomized patients deserve maximum clinical attention after surgery to avoid severe grades of CKD that can lead to ESKD over time, or block oncological treatment in metastatic cases [37]. From a urological and nephrological point of view, this cohort of patients normally undergoes a precise follow-up with different lines of treatment to avoid the progression of renal dysfunction with new drugs to ease the hyperfiltration mechanism [38] and monitor blood hypertension and proteinuria. However, little is known about RC patients’ ideal dietary regimen after nephrectomy because no international guidelines define a clear and shared nutritional strategy. This lack of literature in this onco-nephrological field has to be filled because nutritional status influences cancer, prognosis, quality of life (QoL), and co-morbidities [15]. Therefore, it is crucial to implement interventions to evaluate and enhance these patients’ nutritional status because malnutrition can significantly impact the effectiveness of treatment and the overall outcomes for individuals with RCC [15,20]. Preventing undernourishment is a key target of nutritional interventions for CKD and cancer. Both diseases can result from inadequate intake, metabolic alterations, and inflammation [11]. Wang M. et al. investigated nutritional status in a cohort of 103 patients with advanced kidney cancer, revealing that 76% of subjects were malnourished. In particular, 59 cases (57%) had mild malnutrition and 19 cases (19%) had a severe form. This heightened prevalence can be attributed to the risk factors of patients with advanced kidney cancer, such as age (≥65 years old), body mass index (BMI < 18.5 kg/m^2^), albumin (<40 g/L), hemoglobin (<110 g/L), C-reactive protein (CRP < 3 mg/L), the presence of diabetes, and the presence of anorexia [17]. In this onco-nephrological scenario, our work represents the first worldwide consecutive case series of RC patients with solitary kidneys treated with a combined nephrological and nutritional approach using an LNPHC diet. Our work aimed to solve the “big dilemma” between oncological trends and nephrological perspectives using a methodologic algorithm based on a controlled diet with a “normalized” protein intake to avoid uremic conditions and preserve renal function. By analyzing the 24 h recall diaries, this normalization resulted in a significant reduction in protein intake in adult non-elderly patients, since their usual intake was around 1.2 gr prot/kg daily. Elderly individuals, on the other hand, usually required a different type of normalization, which meant an increase in both daily protein and energy intake. 

This personalized approach is consistent with the concept of precision nutrition, which is a type of precision medicine that focuses on personalized nutrition and dietary adjustments for CKD patients. This approach not only relates to residual renal function, but also considers psychological, functional, and nutritional status, striving towards a more patient-centered nutritional management of kidney health and kidney disease.

This controlled dietary regimen was strictly followed by precise and accurate nephrological and nutritional counseling, blood and urine tests, and the measurement of anthropometric indices. Our results clearly highlighted that the LNPHC diet was able to generate a significant improvement in several nutritional parameters, avoiding malnutrition or catabolism within the cohort. We used different anthropometric indices to investigate this, preferring skinfold thickness measurement and BIA instead of BMI. If, in clinical practice, body mass index is commonly used to diagnose obesity, its accuracy may be influenced by several predictors of muscle mass, such as age, sex, and race. The same goes for the volume overloads that often accompany renal disease. Among patients with CKD, who are often edematous, elderly, and frail, with reduced lean body mass, BMI may not accurately reflect excess body fat. In particular, triceps skinfold is a reliable method for body density assessment (and hence percentage of total body fat) in patients treated with hemodialysis and in patients with severe chronic renal failure [39,40,41]. Given the recent confirmation that total body fat (and not just abdominal fat) participates in increasing the risk of kidney disease, skinfold measurement represents an accurate predictor of body density and an index for managing kidney disease risk [42]. According to the anthropometric measurements taken at baseline, significant differences in fat distribution in terms of triceps fold thickness (*p* < 0.05) and waist circumference (*p* = 0.01) were identified between the post- and pre-intervention stages. Our data showed that, for the same BMI, the values of surrogate measures of abdominal obesity and segmental fat distribution significantly decreased. These data can be further confirmed by focusing on the fat mass index obtained using bioimpedance analysis. This result supports the idea that BMI masks the potential beneficial effects of the dietary treatment on a patient’s nutritional status because it does not make it possible to distinguish between fat and lean mass reliably. In one recent study, some authors suggested using the FM index measured by BIA as the diagnostic index of obesity and demonstrated that obesity was a risk factor for renal function decline and renal function attenuation, as well as a prominent risk factor for the development of kidney cancer [43,44]. In fact, the presence of obesity is linked to the hyperfiltration mechanism in kidneys and results in several biological dysfunctions due to the overload of blood filtration by the glomeruli. The most common features of this phenomenon (well described by Brenner et al. [45]) are related to renal hemodynamic changes, with increases in renal blood flow, sodium reabsorption, the activation of the renin–angiotensin–aldosterone system, the secretion of hormones from the adipose tissue, inflammation, and oxidative stress. Unfortunately, if renal hyperfiltration continues day by day, the inflammation of nephron structures promotes the establishment of fibrosis, resulting over time in tubulum and glomerulus death. In clinical practice, these irreversible pathological modifications lead to the onset of proteinuria and CKD in obese patients. Vivette D’agati [46] et al. investigated the renal effects of obesity in a brilliant work performing renal biopsies in a cohort of obese patients. Their histological evaluation highlighted particular glomerular and tubular damage with glomerulomegalia in the presence or absence of focal and segmental glomerulosclerosis lesions due to the accumulation of lipids. This pathological condition was named “Obesity-Related Glomerulopathy” (ORG), and it resembles diabetic nephropathy in terms of morphological dysfunction, both in the glomerular and tubular compartments. Slowly increasing sub-nephrotic proteinuria is the most frequent presentation of ORG, while massive proteinuria (>5–10 g/day) is detected scarcely.

However, in contrast to diabetic renal disease, obese patients can partially modify the abovementioned glomerulopathy if they lose weight with an appropriate dietary regimen. But obesity is not only related to CKD onset. It is well known that abdominal obesity is associated with a 1.32-fold increased risk of kidney cancer development due to the tissutal inflammation of the kidney derived from the presence of lipids in tubular cells and the interstitium [47]. This metabolic dysfunction is indeed one of the most prominent “primum movens” in cancer establishment, and the increasing worldwide number of renal malignancies among obese patients represents clear feedback on this aspect. Therefore, it is quite intuitive to underline that obesity, CKD, and kidney cancer are strictly connected via common biological pathways able to promote inflammation and fibrosis in a vicious circle.

For these reasons, the reduced FMI measured in our patient cohort through BIA and TSF could be a positive prognostic tool for renal function maintenance. 

Another significant anthropometric factor that we decided to analyze in our work was the FFMI. Fat-free mass is a heterogeneous measure comprising body cell mass, intracellular water, extracellular solids, and extracellular water. VanItallie et al., starting from the concept that muscle mass is a primary component of FFM, hypothesized that the FFMI could act as a simple surrogate indicator when screening for low muscle mass, and proposed the FFMI as an indicator of nutritional status and sarcopenia [48]. The wisdom of this proposal has recently been validated by other authors [49]; they showed that the FFMI had a strong positive correlation with BIA- and DXA-measured appendicular skeletal muscle mass and suggested FFMI cutoff values < 18 kg/m^2^ in men and <15 kg/m^2^ in women for predicting low muscle mass. Our data demonstrated that the patients who did not present low muscle mass before the controlled protein dietary treatment were not affected by a reduction in FFM values, meaning the approach used effectively preserved the patients’ nutritional status.

Moreover, we noticed a stable BCM index and a significant decrease in the ECM/BCM ratio after the LNPHC diet, observations confirming the absence of malnutrition in the cohort. Moreover, these results indicate an improvement in healthy fluid balance. It is worth remembering that ECW and ICW in skeletal muscle reflect the extracellular space filled with plasma and interstitial fluid and the muscle cell mass (i.e., myofibers), respectively [33]. 

The study’s participants did not present severe water imbalances. The ECW/ICW ratio improved significantly when patients followed the dietary treatments, and the TBW % followed an upward trend; the parallel improvement in the FFM and FM ratio can explain this. A comparison study of MAMC and computed tomography demonstrated good consistency in assessing muscle mass in CKD patients and confirmed MAMC as an indicator of muscle consumption [50,51]. Noori et al. demonstrated that MAMC is a better surrogate of DEXA-measured LBM in their validation study [52] and that it may give a good indication of both muscle mass and caloric and protein adequacy [53]. Hence, it can serve as a general index for an appropriate nutritional status; a reduction in MAMC is known as sarcopenia [54] and may be a sign of malnutrition or wasting [53]. We noticed a significant increase in MAMC in our population, and this is an important result to consider when interpreting the overall changes in the anthropometric scenario, indicating both the caloric and protein adequacy of the dietary treatment and muscle mass preservation. 

However, the introduction of the LNPHC diet in patients with an RC solitary kidney did not have such a relevant effect on the nutritional parameters mentioned above. From a nephrological point of view, the significant reduction in urea blood levels after the dietary intervention was dramatically high for all considered patients. This aspect is crucial in terms of CKD treatment over time. In the last decade, several studies have investigated the toxic effects of urea on the development of harmful uremic complications. The most important are related to the development of vascular dysfunction, the accelerated progression of atherosclerosis, damage to the intestinal epithelial barrier with the translocation of bacterial toxins into the bloodstream resulting in systemic inflammation, and finally, renal fibrosis establishment.

Moreover, high levels of plasmatic urea are linked to cognitive impairment, leading to a high rate of dementia onset [55]. This dangerous side effect is related to the capacity of urea to accumulate in the brain, passing the blood–brain barrier and generating inflammation and microvascular damage. Therefore, reducing urea in RC patients can also reduce the risk of cognitive dysfunction over time. It is worth underlining that the decrease in blood urea related to the LNPHC diet is not intuitive in solitary kidneys with RC because the surgical sudden removal of a kidney implicates, in most cases (as also appeared in our study), the development of CKD stage III or more. Therefore, an augmented concentration of nitrogen products of catabolism in the blood of nephropathic patients tends to lead to global renal deficiency. For that very reason, in our cohort of patients, the nutritional intervention with a restriction of protein intake was able to counter the average increase in blood urea due to CKD onset. For all these reasons, the LNPHC diet could play a relevant role in minimizing the accumulation of urea with several potential clinical benefits (primarily slowing the development of a phenotype of renal premature aging).

Regarding serum creatinine and eGFR levels, our study indicates that no statistically significant variations happened in our cohort. This observation is in line with the conclusions of various meta-analyses that have reported conflicting results on the effect of protein diet restriction on glomerular filtration rate in chronic kidney disease [56]. A low protein diet, defined as DPI < 0.8 g/kg/day, has consistently been shown to cause more constriction of the afferent arteriole, resulting in decreased intraglomerular pressure and nephron longevity [36]. On the contrary, different lines of evidence suggest that the ingestion of a high-protein meal leads to an increased glomerular filtration rate (GFR), resulting in ‘glomerular hyperfiltration’ as a result of the amino acid surge, which leads to the dilatation of the ‘afferent’ arteriole and increased intraglomerular pressure. The impact of a diet rich in proteins could be more deleterious in those patients who have a low nephron endowment (as occurring in surgical solitary kidney patients) rather than in individuals with healthy kidney function.

Our work displays some limitations. The first is related to the period: 6 months ± 2. It is plausible that a longer follow-up window could bring out new insights and probably significant ameliorations in the eGFR. The second refers to using an eGFR instead of a measured GFR not influenced by muscle mass or fluid imbalance. The third is derived from the limited number of patients, even though the cohort was very homogenous, with the same type of cancer (RCC), the same surgical technique of radical nephrectomy, and the same post-operative medical follow-up. Finally, our study did not have a control group and was a retrospective study. 

However, to our knowledge, our study is the first in the literature to apply an LNPHC diet in a cohort of RC solitary kidneys, a condition that is in a kind of “clinical limbo” and deserves precise nutritional rules. Until now, guidelines have been totally absent, resulting in a sort of dietary “free will”. 

Further studies are needed to confirm our clinical results.

## 5. Conclusions

The use of a combined nephrological and nutritional approach applying an LNPHC diet is safe for RCC solitary kidney patients. The oncological nutrition dogma on the need for high protein intake has not been confirmed in this cohort of patients; however, this diet with a restricted protein intake was able to significantly avoid malnutrition. Furthermore, it was able to reduce urea blood levels, lowering one of the most important risk factors for a decline in renal function as expected. Indeed, several nutritional parameters were significantly improved while simultaneously avoiding malnutrition and catabolism, despite the low–normal content of proteins, which is normally not advised for oncological patients. It is important to state that these results are only achievable in the context of a combined approach that includes both clinical and nutritional practice. Once again, this important collaboration between nutritionists and nephrologists is placed under the spotlight, since it is essential in the delivery of effective and safe therapeutic strategies which have a special focus on nutrition. 

## Figures and Tables

**Figure 1 nutrients-16-01386-f001:**
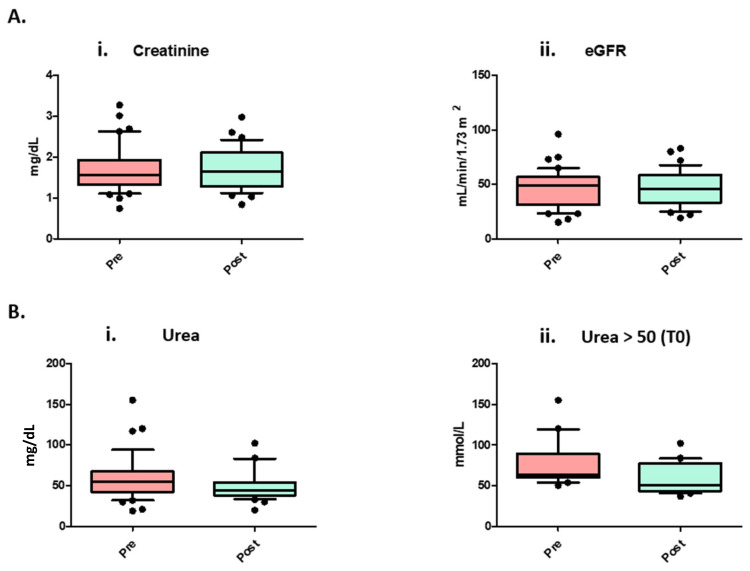
Boxplot of nephrological parameters pre- and post-diet (annotations follow the following format: median [confidence interval]). (**A**) (i) Creatine levels in our population; (**A**) (ii) eGFR calculated with CKD-EPI 2021 formula; (**B**) (i) urea levels in all cohort population; (**B**) (ii) urea levels in the sub-group of patients who had urea > 50 mg/dL at T0. *p*-value was calculated with Kruskal–Wallis sum-rank test.

**Table 1 nutrients-16-01386-t001:** Characteristics of participants at baseline (before treatment). Data are displayed as n (%) and median (range). CKD stage was calculated using the eGFR CKD-EPI (2021) formula. BMI, body mass index; CKD, chronic kidney disease.

Characteristics	Number
Number of patients	40
Age, median (range)	64 (44–86)
Gender, male (%)	32 (80)
Gender, female (%)	8 (20)
BMI (range)	25.8 (23.1–28.7)
Underweight (≤18.5) (%)	0 (0)
Healthy weight (>18.6 ≤ 24.9) (%)	16 (40.0)
Overweight (>24.9 ≤ 30) (%)	16 (40.0)
Obese (>30) (%)	8 (20.0)
Diabetes mellitus (%)	7 (17.5)
Hypertension (%)	22 (55.0)
CKD stage	
CKD-I (%)	1 (2.5)
CKD-II (%)	6 (15.0)
CKD-IIIA (%)	18 (45)
CKD-IIIB (%)	5 (12.5)
CKD-IV (%)	10 (25.0)
CKD-V (%)	0 (0)

**Table 2 nutrients-16-01386-t002:** Anthropometric indices and parameters before and after dietary and nephrological intervention.

Variables	Before Dietary (Visit 1) Median [Range]	After Dietary (Visit 2) Median [Range]	*p*-Value
PA	6.2° [4.1–8.7]	6.3° [4.4–8.8]	0.2530
ECM/BCM	0.81 [0.6–1.4]	0.78 [0.6–1.3]	0.0182 *
BCM/h^2^	12.2 kg/m^2^ [6.9–16.8]	12.5 kg/m^2^ [6.8–16.0]	0.4845
BMI	25.8 kg/m^2^ [19.7–35.4]	25.1 kg/m^2^ [19.6–36.4]	0.0503
Waist Circumference	92.0 cm [74.0–133.0]	90.0 cm [68.0–120.0]	0.0129 *
Waist–Hip Ratio	0.93 [0.68–1.3]	0.91 [0.68–1.1]	0.2464
Skinfold Thickness	15 mm [5.0–32.0]	13 mm [5.0–27.0]	0.0390 *
MAMC	26.3 (23.3–27.6)	26.8 (24.2–28.5)	0.0016 **
ECW/ICW	0.84 [0.57–1.3]	0.82 [0.51–1.1]	0.0178 *
TBW %	59.6 [42.7–67.1]	61.9 [41.2–67.1]	0.0104 *
FM/h^2^	4.7 kg/m^2^ [1.4–14.1]	3.7 kg/m^2^ [1.5–14.9]	0.0044 **
FFM/h^2^	21.2 kg/m^2^ [16.3–27.4]	21.3 kg/m^2^ [13.3–28.2]	0.5960

Annotations follow the following format: mean [confidence interval]. The *p*-value was calculated with the Kruskal–Wallis sum-rank test (statistical significance: * for *p* < 0.05, ** for *p* < 0.01). For simplicity, only the most relevant parameters for our purpose are included in the table. BMI, body mass index; PA, phase angle; ECM, extracellular mass; BCM, body cellular mass; BCM/h^2^, body cellular mass/height square ratio; MAMC, mid-arm muscle circumference; ECW/ICW, extracellular water–intracellular water ratio; TBW, total body water; FM/h^2^, fat mass/height square ratio; FFM/h^2^, free fat mass–height square ratio.

**Table 3 nutrients-16-01386-t003:** Nephrological parameters pre- and post-diet.

	Before Dietary Intervention	After Dietary Intervention	*p* Value
Creatinine	1.6 [0.7–3.3]	1.7 [0.8–3.0]	0.6847
eGFR (EPI 2021)	48.5 [15.0–96.0]	46.0 [19.0–83.0]	0.9812
Urea	55 [19–155]	44 [20–102]	0.0006 **
Urea > 50 mg/dL (To)	64 [50.5–155]	51 [37–102]	0.0001 **

Note: annotations follow the following format: median [confidence interval]. *p*-value was calculated with Kruskal–Wallis sum-rank test (statistical significance: ** for *p* < 0.01).

**Table 4 nutrients-16-01386-t004:** Correlations among BIA parameters and delta eGFR values (from T1 to T0). eGFR is calculated using CKD-EPI 2021 formula.

Correlations							
		Delta CDK-EPI 2021	MAMC T1	ECW/ICW T1	FM/h^2^ T1	ECM/BCM T1	TBW% T1
Delta CDK-EPI 2021	PC	1	−0.020	−0.264	−0.170	−0.216	0.190
	S2T	/	0.911	0.115	0.315	0.205	0.266
MAMC T1	PC	−0.020	1	0.031	0.182	0.222	−0.140
	S2T	0.911	/	0.864	0.302	0.214	0.436
ECW/ICW T1	PC	−0.264	0.031	1	0.523 **	0.855 **	−0.586 **
	S2T	0.115	0.864	/	0.001	0.000	0.000
FM/h^2^ T1	PC	−0.170	0.182	0.523 **	1	0.481 **	−0.960 **
	S2T	0.315	0.302	0.001	/	0.002	0.000
ECM/BCM T1	PC	−0.216	0.222	0.855 **	0.481 **	1	−0.600 **
	S2T	0.205	0.214	0.000	0.002	/	0.000
TBW % T1	PC	0.190	−0.140	−0.586 **	−0.960 **	−0.600 **	1
	S2T	0.266	0.436	0.000	0.000	0.000	/

** Correlation is significant at the 0.01 level (2-tailed).

**Table 5 nutrients-16-01386-t005:** ANCOVA table for the 10% worsening in the eGFR group. F tests the effect of status. This test is based on linearly independent pairwise comparisons among the estimated marginal means.

Univariate Tests							
		Sum of Squares	df	Mean Square	F	Sig.	Partial Eta Squared
Delta ECM/BCM	Contrast	0.003	1	0.003	0.885	0.354	0.028
	Error	0.089	31	0.003			
Delta ECW/ICW	Contrast	0.013	1	0.013	5.174	0.030	0.143
	Error	0.077	31	0.002			

## Data Availability

Research data are unavailable due to privacy or ethical restrictions.

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
