# Peer review of "Effects of a Personalized Diet on Nutritional Status and Renal Function Outcome in Nephrectomized Patients with Renal Cancer"

_nutrients, 2024, doi:10.3390/nu16091386_

Round 1

Reviewer 1 Report

Comments and Suggestions for Authors

This study sought to investigate the efficacy of a Low-Normal Protein High-Calorie (LNPHC) diet in a consecutive cohort of RCC nephrectomized patients using an integrated nephrologist and nutritionist approach. Although this manuscript is great, I raised several concerns regarding methods.

Abstract:

To add more numbers (values in the results section).

Introduction:

It is too short.

What is hypothesis of study?

Methods:

This topic is very weak and did not reflect the data collected.

This study design does make sense. 

What is sample size calculus? Please to add it.

What are methods to assess the body composition? 

What are recommendations to do BIA?

What is food intake pre and after?

What are chronic diseases of patients?

Results:

No new information for correlations among BIA parameters and delta eGFR values.

Discussion:

A deep discussion between renal cancer and GFR and food intake is crucial.

Conclusion:

Is very confusing. The authors discuss the diet, but none data regarding food intake (habits) were provided.

Reviewer 2 Report

Comments and Suggestions for Authors

This distinguished article examines nutritional intervention to preserve renal function after radical nephrectomy, which is indeed a blind spot in nutrition research. Both the study set-up and statistics are more than sound. I have some comments:

- Does a higher risk of osteoporosis exist with this low-protein diet? Especially since oncological patients are already compromised in bone strength.

- Does there exist a difference between left and right nephrectomy (as the latter is much more difficult to perform, yielding more complications)?

Round 2

Reviewer 1 Report

Comments and Suggestions for Authors

Data regarding to food intake is crucial to understanding these findings. 

Author Response

We thank you for your comment. We expanded on the dietary habits of patients in sections 3.1 (lines 188-210) and 4 (lines 346-356). Despite the suggested additions to the manuscript give more insight on the internal evaluation procedure, the topic is not crutial in defining the aim of the study, since our rationale inteds to evaluate the impact of a dietary intervention in patients after surgery. The definition of said intervation is not influenced in any way by the dietary habits of each patient before sugery.